# Replication of machine learning methods to predict treatment outcome with antidepressant medications in patients with major depressive disorder from STAR*D and CAN-BIND-1

**John-Jose Nunez**[1,2], **Teyden T. Nguyen**[2], **Yihan Zhou**[2], **Bo Cao**[3], **Raymond T. Ng**[2], **Jun Chen**[4], **Benicio N. Frey**[5], **Roumen Milev**[6], **Daniel J. Müller**[7], **Susan Rotzinger**[7], **Claudio N. Soares**[5], **Rudolf Uher**[8], **Sidney H. Kennedy**[7], **Raymond W. Lam**[1]*

1 Department of Psychiatry, University of British Columbia, Vancouver, Canada, 2 Department of Computer Science, University of British Columbia, Vancouver, Canada, 3 Department of Psychiatry, University of Alberta, Edmonton, Canada, 4 Shanghai Mental Health Center, Shanghai, China, 5 Department of Psychiatry and Behavioural Neurosciences, McMaster University, Hamilton, Canada, 6 Departments of Psychiatry and Psychology, Queen's University, Kingston, Canada, 7 Department of Psychiatry, University of Toronto, Toronto, Canada, 8 Department of Psychiatry, Dalhousie University, Halifax, Canada

* r.lam@ubc.ca

**Data Availability Statement:** The Python 3 code used to prepare the data, and train and evaluate models, is available in a GitHub repository (https://

## Abstract

### Objectives

Antidepressants are first-line treatments for major depressive disorder (MDD), but 40–60% of patients will not respond, hence, predicting response would be a major clinical advance. Machine learning algorithms hold promise to predict treatment outcomes based on clinical symptoms and episode features. We sought to independently replicate recent machine learning methodology predicting antidepressant outcomes using the Sequenced Treatment Alternatives to Relieve Depression (STAR*D) dataset, and then externally validate these methods to train models using data from the Canadian Biomarker Integration Network in Depression (CAN-BIND-1) dataset.

### Methods

We replicated methodology from Nie et al (2018) using common algorithms based on linear regressions and decision trees to predict treatment-resistant depression (TRD, defined as failing to respond to 2 or more antidepressants) in the STAR*D dataset. We then trained and externally validated models using the clinical features found in both datasets to predict response (≥50% reduction on the Quick Inventory for Depressive Symptomatology, Self-Rated [QIDS-SR]) and remission (endpoint QIDS-SR score ≤5) in the CAN-BIND-1 dataset. We evaluated additional models to investigate how different outcomes and features may affect prediction performance.

github.com/jjnunez11/antidep_pred_replication), which also contains the raw result files. The raw STAR*D data is available from the NDA NIMH study site (29). The raw CAN-BIND-1 data will be available from Brain-CODE, based at the Ontario Brain Institute (https://braininstitute.ca/research-data-sharing/brain-code). The processed data, used to train and evaluate the machine learning models, are also available from the NDA NIMH study site and linked through GitHub. The models used in this paper are available as compressed pickle files, also linked through GitHub.

**Funding:** CAN-BIND is an Integrated Discovery Program carried out in partnership with, and financial support from, the Ontario Brain Institute, an independent non-profit corporation, funded partially by the Ontario government. The opinions, results and conclusions are those of the authors and no endorsement by the Ontario Brain Institute is intended or should be inferred. Additional funding was provided by the Canadian Institutes of Health Research (CIHR), Lundbeck, Bristol-Myers Squibb, Pfizer, and Servier. Funding and/or in kind support is also provided by the investigators' universities and academic institutions. All study medications are independently purchased at wholesale market values. JJN is partially supported by CIHR Team Grant: GACD Mental Health, GAC-154985. CIHR: https://cihr-irsc.gc.ca/e/193.html Lundbeck: https://www.lundbeck.com/ca/en Bristol-Myers Squibb: https://www.bms.com/ Pfizer: https://www.pfizer.ca/ Servier: https://servier.com/en/.

**Competing interests:** JJN, TN, YZ, RTN, JC, and RU have no disclosures. BNF has a research grant from Pfizer. RM has received consulting and speaking honoraria from AbbVie, Allergan, Janssen, KYE, Lundbeck, Otsuka, and Sunovion, and research grants from CAN-BIND, CIHR, Janssen, Lallemand, Lundbeck, Nubiyota, OBI and OMHF. SHK has received honoraria or research funds from Abbott, Alkermes, Allergan, BMS, Brain Canada, CIHR, Janssen, Lundbeck, Lundbeck Institute, Ontario Brain Institute, Ontario Research Fund, Otsuka, Pfizer, Servier, Sunovion, Xian-Janssen, and holds stock in Field Trip Health. RWL has received ad hoc speaking/consulting fees or research grants from: Allergan, Asia-Pacific Economic Cooperation, BC Leading Edge Foundation, Canadian Institutes of Health Research (CIHR), Canadian Network for Mood and Anxiety Treatments (CANMAT), Canadian Psychiatric Association, Healthy Minds Canada, Janssen, Lundbeck, Lundbeck Institute, MITACS, Myriad Neuroscience, Ontario Brain Institute, Otsuka, Pfizer, University Health Network Foundation, and

## Results

Our replicated models predicted TRD in the STAR*D dataset with slightly better balanced accuracy than Nie et al (70%-73% versus 64%-71%, respectively). Prediction performance on our external methodology validation on the CAN-BIND-1 dataset varied depending on outcome; performance was worse for response (best balanced accuracy 65%) compared to remission (77%). Using the smaller set of features found in both datasets generally improved prediction performance when evaluated on the STAR*D dataset.

## Conclusion

We successfully replicated prior work predicting antidepressant treatment outcomes using machine learning methods and clinical data. We found similar prediction performance using these methods on an external database, although prediction of remission was better than prediction of response. Future work is needed to improve prediction performance to be clinically useful.

## Introduction

Depression affects all aspects of life, from impairing social relationships [1], interfering with work functioning [2], and reducing quality of life, to increasing mortality from other medical conditions [3]. Consequently, major depressive disorder (MDD) is the second-leading cause of disability globally [4] and is associated with significant healthcare costs [5].

Clinical guidelines recommend evidence-based treatments such as antidepressant medications for patients with moderate to severe MDD [5]. Unfortunately, only 40–60% of patients will respond to an initial antidepressant and up to a third will show non-response to a second medication [5, 6]. A personalized medicine approach, i.e., predicting whether an individual patient will show response or non-response to an antidepressant, can improve treatment outcomes by providing more intensive treatments sooner in the treatment algorithm. For example, patients predicted to have a high risk of non-response might benefit from starting combination or adjunctive medications instead of monotherapy, or may be prioritized to also engage in psychological treatment.

Prediction of treatment outcome presents a promising application for predictive models trained by machine learning. Briefly, machine learning methods build predictive statistical models from *labelled training* data. These data consist of different types (e.g. age, medication dose) termed *features*. The training data has multiple examples (e.g. subjects in a clinical trial) with these features and each example is labelled with the outcome being predicted (e.g. clinical response). *Cross-validation* (*CV*) is a common method to train and evaluate machine learning models [7]. In CV, the training dataset usually consists of a random sampling of a proportion (e.g., 80%) of the entire dataset, with the remainder (i.e., 20%) held out as *testing dataset*. This *testing dataset* is then used for a validation of the final trained algorithm. Each CV *fold* (e.g. 10-fold) repeats this random sampling of training and testing datasets and then trains and evaluates the model anew. The prediction performance is then measured as the average across all folds. The performance of a specific algorithm will vary based on the data and predictive task and have different trade-offs [8].

A recent systematic review of machine learning to predict treatment outcomes in MDD identified 26 published studies [9]. Of these, 13 used neuroimaging data as

VGH-UBCH Foundation. CAN-BIND is sponsored by non-for-profit and government funding agencies and by commercial sponsors including Lundbeck, Bristol-Myers Squibb, Pfizer, and Servier. None of the sponsors were involved in study design, conduct of the study, data analysis, data sharing (except for the government funded Ontario Brain Institute, as stated in the Data Availability Statement), interpretation of results, drafting of the manuscript, or final approval and publication of the manuscript. This does not alter our adherence to PLOS ONE policies on sharing data and materials.

predictors, three used genetic data such as DNA and micro-RNA, and seven studies used only clinical data (e.g., demographics and symptoms). Predicting treatment response using only readily available clinical data is much more feasible for routine clinical practice compared to neuroimaging or blood testing for genetic and other molecular markers.

Previous studies of clinical data predictors have used a variety of machine learning methods such as elastic net [10], random forests [11], gradient boosting machines (GBM) [12], unsupervised clustering [13], and neural networks [14]. Nie and colleagues [15] compared various predictive and feature selection methods to predict treatment-resistant depression (TRD, defined as not achieving symptom remission after two antidepressant medication trials) in a large dataset from the U.S. Sequenced Treatment Alternatives to Relieve Depression (STAR*D) study [16]. They tested logistic regression with $L_2$ (ridge) and a combination of $L_1$ and $L_2$ regularization (elastic net) [8], random forests, and gradient boosted decision trees. They combined these machine learning algorithms with different methods for selecting features, such as a "clustering-$\chi^2$" method, which uses K-means clustering to prioritize features closest to centroids and then selects the top features based on a $\chi^2$ score [15]. Their analyses resulted in a balanced accuracy of 0.64–0.71 in predicting TRD, depending on the method. Balanced accuracy is an adjusted accuracy which weighs the accuracy for each outcome equally, compensating if one outcome is more common than the other [17].

One well-known limitation of machine learning studies is that predictive modelling results obtained within one dataset may not be accurate when applied to a new and different dataset. This can result from *overfitting*, when a model learns the particularities of a training dataset so well it may be unable to fit new data [8]. Hence, replication studies are very important in machine learning and artificial intelligence [18], particularly given the broader "replication crisis" in science [19], including medicine [20] and health informatics [21]. We are unaware of any prior studies where a set of different authors have replicated work predicting antidepressant treatment outcomes [9, 22].

An independent external validation is the most rigorous method to establish that results from a predictive model will be valid when applied to a different dataset [23, 24], though there are few examples of its use in this field [9, 25]. Decreased performance may result from any changed factor between datasets, such as limited overlapping features or inherent differences in the participants. Nie et al [15] conducted an external validation of their results from the STAR*D dataset by applying their algorithms to a different dataset from another clinical trial (RIS-INT-93). They found that balanced accuracy was worse with the overlapping features in the new dataset, ranging from 0.64–0.67 [15].

We sought to independently replicate the cross-validation findings by Nie et al [15] on the STAR*D dataset, and then externally validate their methodology on another new dataset from the Canadian Biomarker Integration Network in Depression (CAN-BIND)-1 study [26], a multicentre study to discover multi-modal (neuroimaging, electrophysiological, molecular, clinical) biomarkers to predict treatment response in patients with MDD. We chose to replicate the Nie et al study given its relatively strong results, its use of a well-known study that now has its data publicly available, and its evaluation of a variety of common machine learning algorithms. Our hypothesis is that the cross-validation of the STAR*D dataset will replicate the accuracy performance reported by Nie et al [15]. However, we expect the validation on the external CAN-BIND-1 dataset will be less accurate because of the limited number of overlapping features between the two datasets. Our work will also contribute to reproducibility by using automated data processing, publicly accessible code, and the current, open-access version of the STAR*D dataset.

## Methods

### STAR*D dataset

This well described study funded by the U.S. National Institutes of Mental Health (NIMH) sought to investigate next-step treatments of depression after initial agents failed [16]. The study enrolled 4041 outpatients with a DSM-IV [27] diagnosis of MDD without psychotic features (see S1 Table for the inclusion and exclusion criteria) and involved four treatment levels. For level 1, patients initially started citalopram 20–60 mg/d. If they did not achieve remission (score ≤5 on the Quick Inventory of Depressive Symptomatology, Clinician version [QIDS-C] [28]) after 14 weeks, they went on to one of seven different treatment options in level 2, and continued similarly to levels 3 and 4. Patients provided written consent and the study was approved by institutional review boards at the participating institutions. Clinical and demographic data were recorded on enrollment, weeks 2,4,6,9,12, and 14 within each treatment level, and on level exit/entry. We obtained this study's dataset from the publicly available NIMH Data Archive [29], in contrast to the Nie et al [15] study which used an older version of the dataset obtained directly from the STAR*D investigators. We were unable to access this older version of the dataset used by the prior authors, though we know there are some differences such as in feature names and formatting.

### CAN-BIND-1 dataset

This 16-week study conducted across six Canadian academic health centres sought to investigate the prediction of outcomes in patients with DSM-IV MDD treated with medications (S1 Table for the inclusion and exclusion criteria). Full details are described elsewhere [30] but, briefly, 211 patients received eight weeks of open-label treatment with escitalopram 10–20 mg/d, flexibly dosed. Patients who did not respond (defined as a <50% reduction in the Montgomery-Åsberg Depression Rating Scale [MADRS] score] [31] after eight weeks received a further eight weeks of adjunctive aripiprazole 2–10 mg/d, flexibly dosed [32]: responders at eight weeks (≥50% reduction in MADRS score) continued on escitalopram monotherapy for the second eight weeks. Clinical and molecular data were obtained at baseline and at weeks 2, 4, 8, 10 and 16, and imaging data at weeks 2 and 8. All participants provided written consent and institutional review boards at each study site (University of Toronto Health Sciences Research Ethics Board, University of British Columbia Clinical Research Ethics Board, University of Calgary Conjoint Health Research Ethics Board, Queen's University Health Sciences and Affiliated Teaching Hospitals Research Ethics Board, McMaster University Hamilton Integrated Research Ethics Board) provided approval. Table 1 lists information on the participants from both studies.

The two cohorts have similar inclusion and exclusion criteria (S1 Table). Both include participants who meet DSM-IV criteria for MDD, and exclude those with bipolar disorders, significant other psychiatric or substance use disorders, and pregnancy. For comparison to STAR*D level 1, we examined data from the first 8 weeks of CAN-BIND-1.

### Preparation of data

To improve reproducibility, we sought to build an automated pipeline for data cleaning, aggregation, and processing. We built this pipeline in Python 3 using the data handling library pandas [33]. Categorical data were one-hot encoded, and missing data were replaced with a mean, median, or mode, or were computed from other features, depending on the data, further detail is available in the supplemental material.

For the STAR*D datasets, we used all available features from the enrollment, baseline, and week 2 visits of the study. This included symptom scales, demographics, side effects, medical

**Table 1. Characteristics of the cohorts used in the external validation portion of our study, the Sequenced Treatment Alternatives to Relieve Depression (STAR\*D) and Canadian Biomarker Integration Network in Depression (CAN-BIND-1) trials.**

| Characteristic | STAR\*D | | CAN-BIND-1 | |
|---|---|---|---|---|
| | **n** | **%** | **n** | **%** |
| Female:Male | 1858:1151 | 61.7:38.3 | 112:68 | 62.2:37.8 |
| Married/Domestic Partnership | 1266 | 42.1 | 50 | 27.8 |
| Never Married/Divorced/Separated/Widowed | 1743 | 57.9 | 130 | 72.2 |
| Disabled | 488 | 16.2 | 16 | 8.9 |
| Unemployed | 359 | 11.9 | 25 | 13.9 |
| Retired | 122 | 4.1 | 2 | 1.1 |
| Prior depressive episode | 1928 | 64.1 | 107 | 59.4 |
| Substance use Disorder | 209 | 6.9 | 8 | 4.4 |
| Any Anxiety Disorder | 478 | 15.9 | 90 | 50 |
| | Mean | SD | Mean | SD |
| Age | 40.9 | 12.9 | 35.4 | 12.7 |
| Years of education | 13.6 | 3.2 | 13.7 | 2.3 |
| Age in years at onset of depression | 25.3 | 14.3 | 20.8 | 10.1 |
| No. of prior depressive episodes | 3.9 | 8.7 | 2.6 | 2.2 |
| Current episode duration in months | 25.4 | 54 | 26.2 | 33.4 |
| Baseline QIDS-SR Total Score | 15.4 | 4.3 | 16.0 | 4.1 |

QIDS-SR: Quick-Inventory of Depressive Symptomatology, Self Report

and psychiatric history, health care utilization, and scales quantifying functioning at home, school or work. We made additional features as described by Nie et al [15] to represent various depression score subscales and changes in depression scores. As our version of the dataset was not always labeled by the week-of-study, we used days-in-study as a proxy, cross-checked with data that were labelled with both.

To evaluate STAR\*D trained models on the CAN-BIND-1 dataset, only features represented in both datasets could be used. The two studies recorded similar clinical data including symptom scales, demographics, measures of functioning, and history of concurrent psychiatric disease. However, there are some differences between the clinical scales used to quantify these data (Table 2). Notably, the QIDS-C was not done in the CAN-BIND-1 study. However, the patient-rated version, the Quick Inventory of Depressive Symptomatology, Self Rating (QIDS-SR), was completed in both studies. We thus used the QIDS-SR for subject selection and outcome in the external validation analysis. Other overlapping features were selected on exact equivalence, or if conversion was unambiguous, e.g., monthly to yearly income.

**Table 2. Categories of overlapping clinical features, and the scales or forms used to obtain such data from the Sequenced Treatment Alternatives to Relieve Depression (STAR\*D) and Canadian Biomarker Integration Network in Depression (CAN-BIND-1) trials.**

| Overlapping Clinical Features | STAR\*D Source | CAN-BIND-1 Source |
|---|---|---|
| Demographics | Demographics Form | |
| Psychiatric History | Psychiatric Diagnostic Screening Questionnaire [34] | Mini International Neuropsychiatric Interview [35] |
| Functional Impairment | Quality of Life, Enjoyment and Satisfaction Questionnaire–Short Form [36] | |
| | Work and Social Adjustment Scale [37] | Sheehan Disability Scale [38] |
| | Work Productivity & Activity Impairment Questionnaire [39] | Lam Employment Absence and Productivity Scale [40] |
| Depressive Symptoms | Quick Inventory of Depressive Symptomatology, Self report [28] | |

## Subject selection

For the cross-validation STAR*D dataset, we re-implemented the subject selection as described in Nie et al [15]. We included subjects who had any baseline QIDS-C scores, stayed in the study at least four weeks, and either stayed until Level 2 of STAR*D, or left in Level 1 due to achieving remission. After corresponding with the authors of the Nie et al study, we also excluded a small number of subjects who were missing most of their non-QIDS-C data, to better match their dataset.

For our external validation using the CAN-BIND-1 dataset, we adapted the above inclusion criteria. We included subjects with any baseline QIDS-SR scores, who stayed at least four weeks. We again excluded subjects from the STAR*D dataset if they were missing the majority of non-QIDS-C data, or if they were missing baseline QIDS values from the version being used for a model.

## Feature selection

Given the large number of features and correlations, we implemented methods for feature selection. We replicated both methods for feature selection used by Nie et al [15]. We again used elastic net optimized to find around 31 non-zero weights to select those features, replicating Nie et al's attempt to decrease multicollinearity which can impact both interpretability and performance [15, 41]. We also reimplemented their "clustering-$\chi^2$" method, first transposing the data matrix, and then applying k-means clustering. We then picked the features closest to the centroid as representatives of the whole cluster, calculated $\chi^2$ scores, and selected the top 30 features based on these.

## Predictive models

We implemented the methods to create our models as closely as possible to the description in Nie et al [15]. As previously, we used the Python library scikit-learn [42] for the logistic regression, gradient boosted decision trees (GBDT), and random forest models. We implemented all methods from scratch but, since we had some access to their code, when possible we used the hyperparameters from Nie et al [15]. XGBoost [43] was downloaded from its website. As Nie et al [15] did not report how elastic net was implemented, we used scikit-learn's stochastic gradient descent (SGD) model with logistic loss and $L_1/L_2$ regularization [42]. When not provided hyperparameters, we tuned the models using *GridSearchCV* from scikit-learn.

## Training and evaluation

As per Nie et al [15], our training set consisted of 80% of the STAR*D data. While they split the training and validation cohort based on study sites, our data from NDA was not labeled by site, and so we randomly separated the data. We then trained and optimized the models using standard 10-fold cross-validation on this training set [8].

We trained models for the STAR*D cross-validation replication using all features, to predict TRD, as defined as failing to achieve a QIDS-C or QIDS-SR score of five or less in the first two levels of the study. The 20% holdover set was then used to evaluate model performance.

We used separate models trained on the STAR*D data to externally validate performance on the CAN-BIND-1 dataset, using only the overlapping features as previously described. The models were used to predict antidepressant response by eight weeks, as defined as a 50% or greater reduction in their QIDS-SR score. We also used them to predict remission at eight weeks, defined as a QIDS-SR score of five or less. When training on the STAR*D data, the predicted outcomes were the same, but instead we used the first

nine weeks, as STAR*D recorded QIDS scores at week nine instead of at week eight as in the CAN-BIND-1 study.

We further evaluated models with different combinations of targets (TRD, response, or remission), QIDS scale version (clinician or self-report), feature sets, and subject inclusion criteria, to investigate how these changes affect performance. We evaluated these models through cross-validation on the STAR*D data.

We used the method for handling class imbalance as described in Nie et al [15] which utilizes random undersampling [44]. Briefly, models were trained $t$ times, each time using all the data from the minority class, and an equal number of randomly selected data from the majority class. This was repeated $t = 30$ times, with the average of these $t$ models used in ensemble to make the predictions.

We report all results as the mean of repeating the above procedure 100 times to evaluate each model. Numerical differences between our results can be interpreted as statistically significant, as the variance between evaluations is low, which we report alongside p-values from two-tailed t-tests in S5, S8 and S11 Tables. We cannot use McNemar's Test [45] or other methods to directly compare the statistical significance between our results and those of Nie et al [15], as we do not have the specific results of the cross-validation folds used by the original authors. Instead, Table 3 shows the Z-scores for Nie et al's results versus our distribution. We used functions from scikit-learn's metrics module to compute specific metrics such as the AUC [42].

## Results

### Feature and subject selection

Our STAR*D dataset, after following Nie et al's methodology but using the raw data from the NIMH NDA, consists of 480 features. This is fewer than the "approximately 700" Nie

**Table 3. Resulting from replicating a prior study's cross-validation, predicting treatment-resistant depression according to the Quick Inventory of Depressive Symptomatology, Clinician version (QID-C) scale, using data from Sequenced Treatment Alternatives to Relieve Depression.** GBDT: gradient boosting decision tree. Feature selection methods include clustering-$\chi^2$ (30 features) and elastic net (31 features). Results reported as Balanced Accuracy and area-under-curve (AUC). As the replicated study only reported one number for their results, we show the z-score of these against the distribution of our results from 100 runs of 10-fold cross-validation. Additional performance metrics and statistics are documented in S4 and S5 Tables.

| | Balanced Accuracy | | | AUC | | |
|---|---|---|---|---|---|---|
| | Our Result | Nie et al study [15] | Z-Score | Our Result | Nie et al study [15] | Z-Score |
| **Random Forest** | | | | | | |
| Full Features | 73% | 70% | -10 | 0.80 | 0.78 | -18 |
| Clustering-$\chi^2$ | 72% | 68% | -11 | 0.79 | 0.77 | -9 |
| Elastic Net | 72% | 69% | -4 | 0.79 | 0.76 | -5 |
| **GBDT** | | | | | | |
| Full Features | 73% | 70% | -10 | 0.81 | 0.78 | -16 |
| Clustering-$\chi^2$ | 71% | 70% | -3 | 0.78 | 0.77 | -6 |
| Elastic Net | 73% | 70% | -5 | 0.80 | 0.76 | -9 |
| **XGBOOST** | | | | | | |
| Full Features | 71% | 68% | -4 | 0.78 | 0.76 | -4 |
| Clustering-$\chi^2$ | 70% | 67% | -5 | 0.77 | 0.73 | -9 |
| Elastic Net | 70% | 68% | -3 | 0.76 | 0.76 | -0.4 |
| **L$_2$ Logistic Regression** | | | | | | |
| Full Features | 71% | 64% | -16 | 0.78 | 0.69 | -30 |
| Clustering-$\chi^2$ | 72% | 71% | -2 | 0.79 | 0.73 | -34 |
| Elastic Net | 73% | 71% | -3 | 0.80 | 0.77 | -7 |
| **Elastic Net Model** | 70% | 68% | -5 | 0.77 | 0.76 | -2 |

et al reports, though our dataset contains features from all scales noted by these authors, and includes their derived features. The overlapping datasets, consisting of features found in both CAN-BIND-1 and STAR*D, contains 100 features, more than the 22 overlapping features Nie et al report from the external study validation dataset. We document all features in S2 Table.

For the STAR*D datasets, replicating the subject selection from Nie et al [15] for TRD prediction as defined by QIDS-C criteria results in 2520 subjects, with 701 (27.8%) labelled as TRD. These numbers differ slightly from their paper, which reported 2454 subjects with 642 (26.3%) meeting QIDS-C TRD criteria. For the external validation, the STAR*D dataset with QIDS-SR values and overlapping features with the CAN-BIND-1 dataset included 3024 subjects, with 1772 (58.6%) achieving a QIDS-SR response by week 9 and 1295 (42.8%) achieving remission. The CAN-BIND-1 dataset included 180 subjects, with 63 (35.0%) achieving QIDS-SR response by week 8 and 43 (23.9%) achieving remission. Remission and response rates for other targets are shown in S3 Table.

## Replication of cross-validation

Table 3 shows the results of our replication in performing cross-validation on the full STAR*D dataset to predict QIDS-C TRD. We present the results as balanced accuracy, which accounts for class-imbalance, as well as area under the receiver operator characteristic curve (AUC), which evaluates both sensitivity and specificity. We document sensitivity, specificity, F1 and other performance metrics in S4 Table, statistical comparison between these results in S5 Table, and feature importance in S6 Table.

Our models achieved balanced accuracies and AUCs numerically higher than those of Nie et al [15]. The highest balanced accuracy was higher in our study compared to Nie et al (73% versus 71%, respectively). Similarly, our highest AUC was higher at 0.81 versus 0.78, respectively. The z-score of Nie et al's results in our distributions ranges from -0.4 to -34.

We observed a similar benefit by using feature selection methods. As in Nie et al [15], the tree-based methods (Random Forests, GBDT, XGBOOST) generally do not confer a benefit when using the selected features, in contrast to the linear regression methods.

## External validation

Table 4 shows the results of our external validation, training models on the STAR*D dataset with overlapping features and evaluating them on the dataset from CAN-BIND-1. Here, the models are used to predict response (≥50% reduction in QIDS-SR scores) or remission

**Table 4. Performance of our predictive models when trained on the Sequenced Treatment Alternative to Relieve Depression (STAR*D) dataset, and externally evaluated on the Canadian Biomarker Integration Network in Depression (CAN-BIND-1) trial, predicting both response and remission according to the Quick Inventory of Depressive Symptomatology, Self Report Version (QIDS-SR) scale.** See *Methods* for our definition of these outcomes. No feature selection was used before running the models. Additional performance metrics and statistics are documented in S7 and S8 Tables.

| | QIDS-SR Response | | QIDS-SR Remission | |
|---|---|---|---|---|
| | Balanced Accuracy | AUC | Balanced Accuracy | AUC |
| **Random Forest** | 65% | 0.69 | 74% | 0.83 |
| **GBDT** | 63% | 0.70 | 75% | 0.83 |
| **XGBOOST** | 64% | 0.68 | 74% | 0.82 |
| **L$_2$ Logistic Regression** | 61% | 0.65 | 77% | 0.80 |
| **Elastic Net** | 59% | 0.64 | 73% | 0.79 |

GBDT: gradient boosting decision tree, AUC: area-under-curve.

(endpoint QIDS-SR $\leq$ 5). Only overlapping features are used for both training and evaluating the models; 100 such features were available, as summarized in Table 2. Again, models based on decision trees perform better. Our results are higher for predicting QIDS-SR (AUC 0.79–0.83) remission than predicting response (AUC 0.64–0.70). We again provide further performance metrics (S7 Table), statistical comparison (S8 Table), and feature importance (S9 Table).

## Further investigations

To further understand our results, we also compared the performance of response and remission prediction on cross-validation with the STAR*D dataset, as Table 5 shows. We focused on using the Random Forest models without feature selection, given that this was one of the best performing models. Our models continue to predict response worse than they do remission, though the difference is smaller when using QIDS-C instead of QIDS-SR. The supplementary material documents additional metrics (S10 Table), statistical comparison (S11 Table) and feature importance (S12 Table).

We conducted additional cross-validations, again using Random Forests, to investigate whether fewer features could be contributing to the decreased performance of predicting QIDS-SR response external validation (Table 5). On cross-validation, using only the overlapping features between both STAR*D and CAN-BIND-1 increase performance, with balanced accuracy rising to 70% compared to 68% with all STAR*D features. However, we also note that using our feature selection methods to reduce the number of features decreases performance compared to using the full features. Elastic net feature selection drops balanced accuracy to 67%, while clustering-$\chi^2$ lowers it to 65%. Our results for predicting QIDS-SR remission follow a similar pattern on cross-validation (S10 Table), improving when using the overlapping features but not when using feature selection to reduce features. Unlike for QIDS-SR response, QIDS-SR remission results improve when externally validating on CAN-BIND-1, increasing to a balanced accuracy of 74%.

**Table 5. Comparison of model performance with different targets and sets of features, using Random Forests.** Overlapping features are the 100 features in both Canadian Biomarker Integration Network in Depression's CAN-BIND-1's trial and Sequenced Treatment Alternatives to Relieve Depression (STAR*D), while Full uses all 480 features from STAR*D. Clustering-$\chi^2$ Selection (30 features) and Elastic Net Selection (31 features) refer to using these feature selection techniques as defined in *Methods*. Targets include antidepressant response, remission, or treatment-resistant depression (TRD), as defined in *Methods*. Models trained and evaluated using cross-validation (CV) on STAR*D, and we also report again the results of externally validating models on the CAN-BIND-1 dataset after being trained on STAR*D. We report balanced accuracy and area-under-curve (AUC). Additional performance metrics and statistics are documented in S10 and S11 Tables.

| Evaluation | Target | Features | Balanced Accuracy | AUC |
|---|---|---|---|---|
| CV | QIDS-C TRD | Full | 73% | 0.80 |
| CV | QIDS-SR TRD | Full | 74% | 0.83 |
| CV | QIDS-C Remission | Full | 71% | 0.80 |
| CV | QIDS-SR Remission | Full | 72% | 0.81 |
| CV | QIDS-SR Remission | Overlapping | 72% | 0.82 |
| CV | QIDS-C Response | Full | 70% | 0.78 |
| CV | QIDS-SR Response | Full | 68% | 0.76 |
| CV | QIDS-SR Response | Clustering-$\chi^2$ Selection | 65% | 0.72 |
| CV | QIDS-SR Response | Elastic Net Selection | 67% | 0.73 |
| CV | QIDS-SR Response | Overlapping | 70% | 0.78 |
| External Validation | QIDS-SR Remission | Overlapping | 74% | 0.83 |
| External Validation | QIDS-SR Response | Overlapping | 65% | 0.69 |

QIDS: Quick Inventory of Depressive Symptomatology, -SR: Self-Report, -C: Clinician.

## Discussion

Our analyses replicated the cross-validation of the recent Nie et al study using supervised machine learning to predict antidepressant outcomes from clinical data. We then further validated these methods by externally validating them on an unrelated and new dataset. We accomplished this replication and external validation using data processing code, and datasets, that are available to other investigators, to provide a foundation for further replication and improvement. The performance of our replicated prediction is numerically similar, though slightly higher, than that achieved by the prior study. However, the performance for the external validation on our new dataset depended upon which outcome, clinical response or remission, is predicted.

Several possibilities may explain the numerically small improvement in the results of our replicated cross-validation compared to the prior study by Nie et al. We used a newer version of the STAR*D dataset so the datasets have small differences in numbers of subjects and rates of TRD that may lead to small differences in performance. We also used a different method for selecting a hold-out set; because our dataset was not labelled by geographic trial center, we randomly assigned 20% of all subjects to a testing set, in contrast to Nie et al, who used data from three of the fourteen trial centers. This could mean that our training and testing data are more related, leading to overfitting of the machine-learning model. Another possible cause for our increased performance is our full feature set containing fewer features than Nie et al, 480 compared to approximately 700, despite following the data preparation methodology as laid out in their paper and in personal correspondence. This impact may be suggested by the smaller advantage we see in our results that use feature selection methods. Regardless, our slightly improved performance is unlikely to be clinically significant, and we instead interpret our results as a successful replication supporting the findings of Nie et al. and our own implementation of data processing and machine learning methods. To our knowledge, this is the first instance of an independent group replicating prior antidepressant outcome prediction, as prior studies have generally published novel, positive results [9, 25].

For these predictive models to have clinical benefit and be accepted by clinicians [23, 46], they need to have comparable performance when used in different clinical samples. Our external validation results on the CAN-BIND-1 dataset show that the performance can vary based on the specific outcome being measured. When we predict achieving clinical remission using an antidepressant, defined as an endpoint score of five or less on the QIDS-SR scale, we find balanced accuracy performance similar to the cross-validation analysis, with a slight improvement. However, we find reduced performance compared to the cross-validation when predicting clinical response, defined as a 50% or greater reduction on the QIDS-SR scale.

Such large performance differences predicting response versus remission is somewhat unexpected. Prior work has usually not compared performance of these two outcomes on external validation, and the small number of prior studies predicting each of these outcomes have found varied performance with no major differences [9]. Of note, Nie et al [15] found different results, with the prediction of TRD improving when using response criteria compared to remission criteria, such as their Random Forest full feature model having an AUC of 0.78 using QIDS-C remission, but 0.80 with QIDS-C response. As well, on their external validation, when looking at metrics that account for class-imbalance, they find that performance decreases similarly on external validation for both remission and response.

We investigated why response performance dropped on external validation by conducting additional cross-validations on STAR*D datasets. We found that using the 100 features used for the external validation, which are overlapping between STAR*D and CAN-BIND-1 datasets, generally increased performance on both response and remission predictions. This

suggests other factors may be leading to the worsened response prediction on external validation, such as inherent differences in patients between the two datasets. It also provides an example where having to use fewer features due to differences between datasets may actually not hamper performance, but increase it, unlike prior examples–for instance, when Nie et al only used overlapping features on STAR*D cross-validation, they noted performance decreases. This is likely related to the number and types of features; our external validation dataset had more features overlapping than did this prior work; 100 compared to 22.

Together, these results have implications for future work and clinical deployment. While prior work has often found little difference between predicting antidepressant outcomes, we show that the same machine learning methodology can lead to different performance when predicting response and remission, including when externally validated. Our improved results when using the overlapping features has implications for deployment in clinical settings, where it may not be feasible to collect all the information required to replicate all features from a clinical trial, and for further external validation, transfer learning, or other applications where the number of overlapping features between datasets may be limited.

While our results are generally an improvement over Nie et al, further performance increase is likely needed for clinical applicability. Our results may also have implications for how to improve these predictive models. We note that the two feature selection methods attempted often worsened cross-validation performance. This may suggest that alternative techniques and parameters for feature selection may lead to increased performance. Whether it be with different feature selection techniques, different models, or other changes, those seeking to improve the performance of predicting antidepressant effect will be able to easily reproduce our work, using the same dataset. This will allow better understanding of how possible changes affect performance.

## Limitations

There are limitations to our cross-validation replication with Nie et al, including using a different version of the STAR*D dataset and a different method of data processing. Our cross-validation evaluation could not be based on study location, as our dataset does not include this information. Similarly, our deployment of the machine learning analyses may have differences, despite our best efforts, such as our different number of features in the full STAR*D dataset. Future work can support replicability by ensuring their data and methods are publicly available.

The results of our external validation may largely depend on the specific differences and similarities between the STAR*D and CAN-BIND-1 datasets. It is unclear how broadly these findings apply, and future work could investigate this by repeating our CAN-BIND-1 dataset analysis with a different training dataset. This may also help elucidate further factors impacting the difference in performance of remission and response predictions.

## Conclusions

Replication and external validation may play an important role in driving clinician acceptance and applicability of machine learning methods in psychiatry. Our results represent the first independent replication of prior work using machine learning models to predict antidepressant outcomes in non-psychotic MDD. We successfully replicate the prior Nie et al cross-validation results using a newer version of their data that is publicly available. Our external validation of these methods on a new, independent dataset from a different country found that performance was similar to cross-validation when predicting clinical remission, although performance was reduced when predicting clinical response. These results motivate future work

to investigate the generalizability of this finding, as well as other efforts to improve prediction performance. Our work facilitates future research by using reproducible and publicly available data and methodology.

## Supporting information

**S1 Table. Inclusion and exclusion criteria for the two studies, CAN-BIND-1 and STAR*D.**
(DOCX)

**S2 Table. Features included in our datasets, and methods used for filling missing data.**
(XLSX)

**S3 Table. Number of subjects in datasets and their rates of predicted outcomes.**
(XLSX)

**S4 Table. Additional statistics from replicating a prior study's cross-validation predicting TRD according to the QIDS-C scale using data from STAR*D.**
(XLSX)

**S5 Table. Statistical comparison of results from replicating a prior study's cross-validation predicting TRD according to the QIDS-C scale using data from STAR*D.**
(CSV)

**S6 Table. Top 31 features by feature importance when replicating a prior study's cross-validation predicting TRD according to the QIDS-C scale using data from STAR*D.**
(CSV)

**S7 Table. Additional statistics when our models are trained on STAR*D, and externally evaluated on CAN-BIND-1, predicting both response and remission according to QIDS-SR.**
(XLSX)

**S8 Table. Statistical comparison of results from our predictive models when trained on STAR*D, and externally evaluated on CAN-BIND-1, predicting both response and remission according to QIDS-SR.**
(CSV)

**S9 Table. Top 31 features by feature importance of our predictive models when trained on STAR*D, and externally evaluated on CAN-BIND-1, predicting both response and remission according to QIDS-SR.**
(CSV)

**S10 Table. Additional statistics from our comparison of model performance with different targets and sets of features, using Random Forests.**
(XLSX)

**S11 Table. Statistical comparison of results from our comparison of model performance with different targets and sets of features, using Random Forests.**
(CSV)

**S12 Table. Top 31 features by feature importance from our comparison of model performance with different targets and sets of features, using Random Forests.**
(CSV)

## Acknowledgments

We acknowledge the contributions of the CAN-BIND Investigator Team (https://www.canbind.ca/about-can-bind/our-team/full-can-bind-team/).

**NIMH NDA Disclaimer:** Data and/or research tools used in the preparation of this manuscript were obtained from the National Institute of Mental Health (NIMH) Data Archive (NDA). NDA is a collaborative informatics system created by the National Institutes of Health to provide a national resource to support and accelerate research in mental health. Dataset identifier(s): *[NIMH Data Archive Collection ID(s) or NIMH Data Archive Digital Object Identifier (DOI)]*. This manuscript reflects the views of the authors and may not reflect the opinions or views of the NIH or of the Submitters submitting original data to NDA.

## Author Contributions

**Conceptualization:** John-Jose Nunez, Teyden T. Nguyen, Yihan Zhou, Bo Cao, Raymond T. Ng, Roumen Milev, Daniel J. Müller, Susan Rotzinger, Claudio N. Soares, Rudolf Uher, Sidney H. Kennedy, Raymond W. Lam.

**Data curation:** John-Jose Nunez, Teyden T. Nguyen, Yihan Zhou.

**Formal analysis:** John-Jose Nunez, Raymond W. Lam.

**Funding acquisition:** Sidney H. Kennedy, Raymond W. Lam.

**Investigation:** John-Jose Nunez, Teyden T. Nguyen, Yihan Zhou, Raymond W. Lam.

**Methodology:** John-Jose Nunez, Teyden T. Nguyen, Yihan Zhou, Bo Cao, Raymond T. Ng, Rudolf Uher, Raymond W. Lam.

**Project administration:** Jun Chen, Benicio N. Frey, Roumen Milev, Daniel J. Müller, Susan Rotzinger, Claudio N. Soares, Sidney H. Kennedy, Raymond W. Lam.

**Resources:** John-Jose Nunez, Raymond W. Lam.

**Software:** John-Jose Nunez, Teyden T. Nguyen, Yihan Zhou, Raymond T. Ng.

**Supervision:** Bo Cao, Raymond T. Ng, Rudolf Uher, Raymond W. Lam.

**Validation:** John-Jose Nunez, Teyden T. Nguyen.

**Visualization:** Bo Cao.

**Writing – original draft:** John-Jose Nunez.

**Writing – review & editing:** Bo Cao, Raymond T. Ng, Jun Chen, Benicio N. Frey, Roumen Milev, Daniel J. Müller, Susan Rotzinger, Claudio N. Soares, Rudolf Uher, Sidney H. Kennedy, Raymond W. Lam.

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
