## [Decision Letter · Decision Letter 0]

7 Apr 2021

PONE-D-21-05237

Replication of Machine Learning Methods to Predict Treatment Outcome with Antidepressant Medications in Patients with Major Depressive Disorder from STAR*D and CAN-BIND-1.

PLOS ONE

Dear Dr. Lam,

Thank you for submitting your manuscript to PLOS ONE. After careful consideration, we feel that it has merit but does not fully meet PLOS ONE’s publication criteria as it currently stands. Therefore, we invite you to submit a revised version of the manuscript that addresses the points raised during the review process.

We look forward to receiving your revised manuscript.

Kind regards,

Zezhi Li, Ph.D., M.D.

Academic Editor

PLOS ONE

Journal Requirements:

2. Thank you for including your ethics statement:  "All participants provided written consent and institutional review boards provided approval. ".   

"JN, TN, YZ, RTN, JC, and RU have no competing interests.

BNF has a research grant from Pfizer.

RM has received consulting and speaking honoraria from AbbVie, Allergan, Janssen, KYE, Lundbeck, Otsuka, and Sunovion, and research grants from CAN-BIND, CIHR, Janssen, Lallemand, Lundbeck, Nubiyota, OBI and OMHF.

SHK has received honoraria or research funds from Abbott, Alkermes, Allergan, BMS, Brain Canada, CIHR, Janssen, Lundbeck, Lundbeck Institute, Ontario Brain Institute, Ontario Research Fund, Otsuka, Pfizer, Servier, Sunovion, Xian-Janssen, and holds stock in Field Trip Health.

RWL has received ad hoc speaking/consulting fees or research grants from: Allergan, Asia-Pacific Economic Cooperation, BC Leading Edge Foundation, Canadian Institutes of Health Research (CIHR), Canadian Network for Mood and Anxiety Treatments (CANMAT), Canadian Psychiatric Association, Healthy Minds Canada, Janssen, Lundbeck, Lundbeck Institute, MITACS, Myriad Neuroscience, Ontario Brain Institute, Otsuka, Pfizer, University Health Network Foundation, and VGH-UBCH Foundation. "

We note that you received funding from a commercial source:Lundbeck, Bristol-Myers Squibb, Pfizer, and Servier.

5. One of the noted authors is a group or consortium [CAN-BIND Investigator Team]. In addition to naming the author group, please list the individual authors and affiliations within this group in the acknowledgments section of your manuscript. Please also indicate clearly a lead author for this group along with a contact email address.

Reviewers' comments:

Reviewer's Responses to Questions

**Comments to the Author**

1. Is the manuscript technically sound, and do the data support the conclusions?

Reviewer #1: Partly

Reviewer #2: Yes

2. Has the statistical analysis been performed appropriately and rigorously? 

Reviewer #1: N/A

Reviewer #2: Yes

3. Have the authors made all data underlying the findings in their manuscript fully available?

Reviewer #1: No

Reviewer #2: Yes

4. Is the manuscript presented in an intelligible fashion and written in standard English?

Reviewer #1: Yes

Reviewer #2: Yes

5. Review Comments to the Author

Reviewer #1: This paper replicates a machine learning model related to Major Depressive Disorder, gets consistent performance on the same dataset compared to the original study, and evaluates the model on an external dataset. Please see my comments below.

The performance dropped by ~10% when applied the model to the external dataset. The study claims that different feature selection methods may improve the performance on the external dataset and presents the results on Table 5. However, this basically means the model is actually tuned on the external dataset, which might make the results biased. By tuning the models and validating on the external dataset multiple times, the external dataset is no longer 'external' anymore. I would suggest the following: (1) split the original dataset into N folds; (2) train the model using the full features using N-1 folds and evaluate on the remaining fold, repeating N times; (3) repeat (2) using different feature selection methods as proposed in the study; (4) comparing the mean and standard deviation between (2) and (3) to firstly show the feature selection improves the performance on the original dataset. If so, apply these models to the external set and compare mean and standard deviation again.

Also, please also include specificity, sensitivity, and F1-score. The current two metrics are not enough to show the full spectrum of the performance.

Further, the study should explain in detail why chose this model to replicate. In general, there are many more advanced models can be applied to this problem. Not sure why chose to replicate this specific model instead of developing a novel model.

Reviewer #2: Manuscript Summary:

The authors successfully replicated the previous work of Nie et al that predicted antidepressant treatment outcomes using machine learning methods and clinical data. They used a newer version of the STAR*D dataset and a different method for selecting a hold-out set. The results showed slightly better balanced accuracy than that from Nie et al. The external validation on the CAN-BIND-1 dataset and the feature selection were performed, and they claim that the smaller set of features generally improved prediction performance.

Some comments:

1. Although this manuscript successfully repeated previous studies and improved the model performances, the innovation of the manuscription is a bit limited. Please provide the significance and more advantages of this project.

2. Please consider adding the number of features set based on Clustering-χ2 and elastic net methods in the cross-validation results (table 3), if they are available

3. AUC is generally used to evaluate the performance of a binary classification model. In this manuscript, please consider adding more description on how to calculate the AUC values in the methods section.

4. The link about Python 3 code in the GitHub repository should be provided in the Data Availability section.

5. Please briefly introduce the differences of STAR*D database between the previous article (Nie et al) and this manuscript.

6. PLOS authors have the option to publish the peer review history of their article (what does this mean?). If published, this will include your full peer review and any attached files.

Reviewer #1: No

Reviewer #2: No

---

## [Author Response · Author response to Decision Letter 0]

6 May 2021

Please see Response to Reviewers document

---

## [Decision Letter · Decision Letter 1]

27 May 2021

Replication of Machine Learning Methods to Predict Treatment Outcome with Antidepressant Medications in Patients with Major Depressive Disorder from STAR*D and CAN-BIND-1.

PONE-D-21-05237R1

Dear Dr. Lam,

We’re pleased to inform you that your manuscript has been judged scientifically suitable for publication and will be formally accepted for publication once it meets all outstanding technical requirements.

Kind regards,

Zezhi Li, Ph.D., M.D.

Academic Editor

PLOS ONE

Additional Editor Comments (optional):

Reviewers' comments:

Reviewer's Responses to Questions

**Comments to the Author**

1. If the authors have adequately addressed your comments raised in a previous round of review and you feel that this manuscript is now acceptable for publication, you may indicate that here to bypass the “Comments to the Author” section, enter your conflict of interest statement in the “Confidential to Editor” section, and submit your "Accept" recommendation.

Reviewer #1: All comments have been addressed

Reviewer #2: All comments have been addressed. 

2. Is the manuscript technically sound, and do the data support the conclusions?

Reviewer #1: Yes

Reviewer #2: (No Response)

3. Has the statistical analysis been performed appropriately and rigorously? 

Reviewer #1: N/A

Reviewer #2: (No Response)

4. Have the authors made all data underlying the findings in their manuscript fully available?

Reviewer #1: No

Reviewer #2: (No Response)

5. Is the manuscript presented in an intelligible fashion and written in standard English?

Reviewer #1: Yes

Reviewer #2: (No Response)

6. Review Comments to the Author

Reviewer #1: My comments have been fully addressed and I do not have additional comments on this manuscript. Thanks the authors for substantially revising it.

Reviewer #2: (No Response)

7. PLOS authors have the option to publish the peer review history of their article (what does this mean?). If published, this will include your full peer review and any attached files.

Reviewer #1: No

Reviewer #2: No

---

## [Editor Report · Acceptance letter]

17 Jun 2021

PONE-D-21-05237R1 

Replication of Machine Learning Methods to Predict Treatment Outcome
with Antidepressant Medications in Patients with Major Depressive Disorder
from STAR*D and CAN-BIND-1 

Dear Dr. Lam:

I'm pleased to inform you that your manuscript has been deemed suitable for publication in PLOS ONE. Congratulations! Your manuscript is now with our production department. 

Kind regards, 

on behalf of

Dr. Zezhi Li 

Academic Editor

PLOS ONE